statistical physics/statistics/cognition

language regularities, Heaps' Law, tagged texts, grammatical classes, statistical anomalies

**Author for correspondence:**
D. H. Zanette
e-mail: zanette@cab.cnea.gov.ar

# Heaps' Law and Heaps functions in tagged texts: evidences of their linguistic relevance

## A. Chacoma[1] and D. H. Zanette[2]

[1]Instituto de Física Enrique Gaviola, Consejo Nacional de Investigaciones Científicas y Técnicas and Universidad Nacional de Córdoba, Ciudad Universitaria, 5000 Córdoba, Pcia. de Córdoba, Argentina
[2]Centro Atómico Bariloche and Instituto Balseiro, Comisión Nacional de Energía Atómica and Universidad Nacional de Cuyo, Consejo Nacional de Investigaciones Científicas y Técnicas, Av. Bustillo 9500, 8400 San Carlos de Bariloche, Pcia. de Río Negro, Argentina

DHZ, 0000-0003-0681-0592

We study the relationship between vocabulary size and text length in a corpus of 75 literary works in English, authored by six writers, distinguishing between the contributions of three grammatical classes (or 'tags,' namely, *nouns*, *verbs* and *others*), and analyse the progressive appearance of new words of each tag along each individual text. We find that, as prescribed by Heaps' Law, vocabulary sizes and text lengths follow a well-defined power-law relation. Meanwhile, the appearance of new words in each text does not obey a power law, and is on the whole well described by the average of random shufflings of the text. Deviations from this average, however, are statistically significant and show systematic trends across the corpus. Specifically, we find that the appearance of new words along each text is predominantly retarded with respect to the average of random shufflings. Moreover, different tags add systematically distinct contributions to this tendency, with *verbs* and *others* being respectively more and less retarded than the mean trend, and *nouns* following instead the overall mean. These statistical systematicities are likely to point to the existence of linguistically relevant information stored in the different variants of Heaps' Law, a feature that is still in need of extensive assessment.

## 1. Introduction

Among the handful of statistical regularities reported for written human language during the last several decades [1,2], Zipf's Law [3,4] and Heaps' Law [5,6] are undoubtedly the best

known and most thoroughly studied [7]. Zipf's Law establishes a quantitative connection between the number of occurrences of a given word in a corpus of text, $m$, and the rank of that word in a list where all the different words in the corpus are ordered by their decreasing frequency, $r$. According to Zipf's Law, within a wide range of values of $r$, a power-law relation $m \propto r^{-z}$ is verified. From the analysis of a broad variety of corpora, it has been empirically shown that this relation holds in many languages, with an exponent usually close to unity, $z \approx 1$.

Heaps' Law, in turn, postulates a power-law relation of the form $V \propto N^h$ between the size a corpus measured in words—namely, the number of *word tokens*, $N$—and the vocabulary size—the number of *word types*, $V$. The exponent $h$ is positive and lower than unity, which accounts for the fact that the vocabulary grows slower than the corpus itself.

The mathematical-statistical connection between Zipf's Law and Heaps' Law has been discussed by several authors [8–11], whose main goal has been to prove the formal link between the power-law relations postulated by the two laws. However, even a superficial inspection of various reports on the relationship between the sizes of actual corpora and their vocabularies reveals systematic deviations from a power-law interdependence [9,10,12–15], a fact that has been dealt with only rarely [16,17].

To be precise, Heaps' Law (and, similarly, Zipf's Law) can be formulated in at least three variants, depending on the nature of the corpus being analysed. Some of the longest corpora for which Heaps' Law has been studied correspond to concatenations of many individual texts, such as the digitized documents of Project Gutenberg [15,18] or Google Books [19,20], or the novels of certain authors [14]. In this case, the whole corpus is divided into sections following some prescribed criterion, and the values of $V$ and $N$ are recorded for each section. Very recently, concatenated speech utterances have also been analysed from this perspective [21]. In the second variant, a corpus is formed by several individual texts which share a common attribute—for instance, the Wikipedia articles written in English [1], or the academic papers on a certain subject published in a given period [22]— and the values of $V$ and $N$ are those which correspond to each text. Finally, taking a corpus formed by a single text with a total of $N$ word tokens and $V$ word types, the progressively growing sizes of text and vocabulary can be related to each other as the text develops from beginning to end [9,12,13,23]. At each step $n$ along the text, the number of words types $v(n)$ used until then is recorded, and the function $v(n)$—starting at $v(1) = 1$ and ending at $v(N) = V$—characterizes Heaps' Law in this third variant.

Due to the kinds of corpora used in the first two variants, such formulations of Heaps' Law bear information on global features of language, related to the overall number of different words needed to produce a text corpus of a certain (typically large) size, complying with the rules of grammar but not necessarily self-consistent with respect to its semantic contents. The third variant, in contrast, records the progressive incorporation of new words as a text—presumably coherent in subject, style, and genre—builds up. In fact, the function $v(n)$ keeps record, in simplified mathematical terms, of the succession of decisions made by the author of the text, who either uses already employed words or adds new ones in response to both linguistic principles and the purpose of constructing long-term meaning and context. In this sense, such approach is expected to bear valuable quantitative information on how language works when a consistent written discourse is being produced.

In the present contribution, we analyse Heaps' Law for a collection of 75 literary texts written in English by six British and North American novelists from the nineteenth and twentieth centuries (details on this corpus are given in §2). The main novelty in our analysis is that we use a tagged version of each text, discriminating between three classes of words (*nouns*, *verbs* and *others*) on the basis of their grammatical function. This allows us to discern between the contribution of each class to the building-up of the vocabulary. Beginning by the above second variant of Heaps' Law, we show in §3 that the total number of word tokens and types for the 75 works in the corpus collectively obey a well-defined power-law relation. Each grammatical class by itself, moreover, satisfies a similar relation. In §4, we study the *Heaps functions* $v(n)$ constructed for each individual text, as explained above for the third variant of Heaps' Law. By quantifying its difference with the average Heaps function of all the random shufflings of the text—whose analytical form does *not* follow a power law—we demonstrate that $v(n)$ generally possesses a high statistical significance, that might be related to relevant linguistic features associated with discourse production. Finally, in §5, we show that the three grammatical classes contribute very differently to the difference between $v(n)$ and the shuffled-text Heaps function, adding another evidence of the linguistic relevance of Heaps' Law. Our results are briefly discussed in the last section.

## 2. Material and methods

The corpus analysed in this contribution consists of 75 English texts in narrative style—spanning from fables, tales, and short stories to full novels—written by six well-known British and North American authors: J. Austen, Ch. Dickens, A. Huxley, E. A. Poe, M. Twain, and H. G. Wells. The dates of their first publication cover a period of some 150 years, between the decades of 1810 and 1960. The number of word tokens vary from $N \approx 800$ (with a vocabulary of $V \approx 300$ word types) to $N \approx 350\,000$ (with $V \approx 22\,000$). Table 1 gives the complete list of works, indicating author, identification code (to be used along the paper), title, publication year, length and vocabulary size. Digitized versions of the 75 works were obtained as plain-text files from the repositories at Project Gutenberg [18] and Faded Page [24].

Each file obtained from these repositories was first manually cleaned up of contents not belonging to the original text. This included editorial preambles and closing notes, transcription attributions, page numbering and figure captions among a few others. The entire cleaned texts were then automatically processed using the Natural Language Toolkit library (NLTK) in Python [25,26]. This allowed, first, to tokenize each text into single words, punctuation marks and other separators. Punctuation marks were preserved in the tokenized text, as they are crucial elements in the syntax analysis used at the ensuing stage of word tagging. On the other hand, capitalization was ignored. The stage of tagging consisted of classifying all the words into the 35 lexical categories recognized by the NLTK's POS (part of speech) tagger. Tagging combines a collection of techniques, ranging from dictionary search to hidden Markov models training [27]. In our analysis, in order to make the classification more intelligible—and, at the same time, to increase the statistical weight of sampling—we have grouped the 35 categories into three grammatical classes: *nouns*, including the categories of common and proper nouns in singular and plural, as well as personal pronouns; *verbs*, including the categories of verbs in all persons and tenses; and *others*, comprising all the remaining categories. To simplify the nomenclature, we refer to each one of these three classes as a 'tag.' For each file, the output of the whole process was a sequence of words corresponding to the original text, each word added with its tag attribute.

Other specific analytical and numerical procedures are opportunely described along the paper.

## 3. Assessing Heaps' Law across the corpus

We start the analysis by studying the relation between the total vocabulary $V$ and the total length $N$ for the 75 works in the corpus, as recorded in table 1. This corresponds to the second variant of Heaps' Law referred to in the Introduction. The main panel of figure 1 shows a log-log plot of $V$ versus $N$, with different symbols for each author. The straight line is a linear fit, corresponding to a power-law dependence, $V \propto N^h$, with $h = 0.68 \pm 0.01$. Pearson's correlation coefficient is $r = 0.99$, indicating very good agreement with Heaps' Law. Moreover, the value found for the exponent $h$ lies within the interval of values reported for similar corpora [1,14,16].

Note that, regarding the works of different authors, there are a few systematic deviations with respect to the power law fitted for the whole corpus. Austen's novels (aus), in particular, possess a relatively small vocabulary in relation to their length. Her longest works, with $N > 10^5$, have vocabularies which are two-thirds as rich as expected on the average. By contrast, Huxley's works (hux) lie systematically above the fitting, indicating relatively abundant vocabularies. For his longest novels, the value of $V$ is between 20 and 30% above the average. The inset in figure 1 shows the same data as in the main panel but in linear scales, to facilitate appraising these differences.

Figure 2a shows the number of word tokens in each tagged class (tag = *nouns*, *verbs* and *others*) as a function of the total number of tokens in each text. The straight lines in this log-log plot have unitary slope, clearly showing that each tag represents a well-defined fraction of the total length, $N_{\text{tag}} \approx \alpha_{\text{tag}}N$. Specifically, we find $\alpha_n = 0.313 \pm 0.002$, $\alpha_v = 0.186 \pm 0.001$ and $\alpha_o = 0.501 \pm 0.002$, for *nouns*, *verbs* and *others*, respectively. The values of $\alpha_n$ and $\alpha_v$ are in reasonable agreement with those reported by standard sources [28], although accounts for extensive corpora are still rare.

Regarding the fraction of each tag in the vocabularies, figure 2b shows that approximate proportionality, $V_{\text{tag}} \approx \beta_{\text{tag}}V$, holds when the vocabulary is large ($V \gtrsim 4000$), with $\beta_n = 0.47 \pm 0.01$, $\beta_v = 0.28 \pm 0.01$, and $\beta_o = 0.247 \pm 0.004$. Note that the tag *others*, whose overall frequency over the whole texts is above 50%, represents the smallest fraction in large vocabularies, with less than 25%. For smaller vocabularies, on the other hand, the relative quantity of words in each tag changes, with *others* becoming more abundant as the vocabulary size decreases.

**Table 1.** List of works in the present corpus. When two years are indicated in the second column (aus07, wel04) the first one corresponds to the (estimated) year of writing. The two last columns give the length $N$ (number of word tokens) and the vocabulary size $V$ (number of word types).

| author and code | title (publication year) | $N$ | $V$ |
| --- | --- | --- | --- |
| J. Austen | | | |
| aus01 | Pride and Prejudice (1813) | 12 2576 | 8698 |
| aus02 | Emma (1815) | 161 338 | 10 241 |
| aus03 | Sense and Sensibility (1811) | 120 373 | 8631 |
| aus04 | Northanger Abbey (1817) | 77 937 | 7822 |
| aus05 | Persuasion (1818) | 83 821 | 7553 |
| aus06 | Mansfield Park (1814) | 160 770 | 10 883 |
| aus07 | Lady Susan (1794/1871) | 23 254 | 3495 |
| Ch. Dickens | | | |
| dic01 | Oliver Twist (1838) | 159 565 | 14 851 |
| dic02 | A Christmas Carol (1843) | 28 954 | 5215 |
| dic03 | The Cricket on the Hearth (1845) | 31 440 | 5818 |
| dic04 | The Haunted Man and the Ghost's Bargain (1848) | 33 778 | 5818 |
| dic05 | Hard Times (1854) | 102 977 | 13 086 |
| dic06 | A Tale of Two Cities (1859) | 137 153 | 14 040 |
| dic07 | Great Expectations (1860) | 187 455 | 15 717 |
| dic08 | The Mystery of Edwin Drood (1870) | 95 252 | 12 135 |
| dic09 | David Copperfield (1850) | 356 161 | 22 486 |
| dic10 | The Pickwick Papers (1836) | 300 495 | 24 016 |
| dic11 | Little Dorrit (1857) | 38 553 | 23 311 |
| dic12 | Barnaby Rudge (1841) | 255 447 | 20 158 |
| dic13 | The Chimes (1844) | 30 570 | 5822 |
| A. Huxley | | | |
| hux01 | The Tilloston Banquet (1922) | 14 393 | 3534 |
| hux02 | Antic Hay (1923) | 87 974 | 13 908 |
| hux03 | Chrome Yellow (1921) | 57 208 | 10 342 |
| hux04 | Farcical History of Richard Greenow (1920) | 20 478 | 4954 |
| hux05 | Those Barren Leaves (1925) | 122 484 | 16 807 |
| hux06 | Brave New World (1932) | 63 778 | 11 078 |
| hux07 | Eyeless in Gaza (1936) | 146 216 | 19 068 |
| hux08 | The Devils of Loudun (1952) | 124 116 | 17 282 |
| hux09 | Island (1962) | 107 723 | 15 845 |
| hux10 | Happily Ever After (1920) | 13 704 | 3283 |
| hux11 | Eupompus Gave Flavor to Art by Numbers (1920) | 3334 | 1225 |
| hux12 | Cynthia (1920) | 2437 | 935 |
| hux13 | The Bookshop (1920) | 1698 | 776 |
| hux14 | The Death of Lully (1920) | 4455 | 1443 |
| hux15 | The Gioconda Smile (1921) | 11 190 | 2756 |
| E. A. Poe | | | |
| poe01 | The Purloined Letter (1844) | 7042 | 1950 |

(*Continued.*)

| author and code | title (publication year) | N | V |
|---|---|---|---|
| poe02 | The Thousand-and-Second Tale of Scheherazade (1845) | 5660 | 1737 |
| poe03 | A Descent into the Maelström (1841) | 7035 | 1878 |
| poe04 | Von Kempelen and his Discovery (1849) | 2783 | 993 |
| poe05 | Mesmeric Revelation (1844) | 3742 | 1133 |
| poe06 | The Facts in the Case of M. Valdemar (1845) | 3559 | 1177 |
| poe07 | The Black Cat (1843) | 3925 | 1348 |
| poe08 | The Fall of the House of Usher (1839) | 7186 | 2234 |
| poe09 | Silence-a Fable (1838) | 1359 | 427 |
| poe10 | The Masque of the Red Death (1842) | 2425 | 900 |
| poe11 | The Cask of Amontillado (1846) | 2341 | 850 |
| poe12 | The Imp of the Perverse (1845) | 2437 | 936 |
| poe13 | The Island of the Fay (1841) | 1974 | 823 |
| poe14 | The Assignation (1834) | 4473 | 1613 |
| poe15 | The Pit and the Pendulum (1842) | 6152 | 1788 |
| M. Twain | | | |
| twa01 | The Gilded Age (1873) | 162 003 | 16 879 |
| twa02 | The Prince and the Pauper (1881) | 69 693 | 10 869 |
| twa03 | A Connecticut Yankee in King Arthur's court (1889) | 119 560 | 14 200 |
| twa04 | The American Claimant (1892) | 65 776 | 9462 |
| twa05 | The Tragedy of Pudd'nhead Wilson (1893) | 53 274 | 8175 |
| twa06 | Personal Recollections of Joan of Arc (1896) | 151 693 | 14 697 |
| twa07 | A Horse's Tale (1907) | 17 127 | 3906 |
| twa08 | The Mysterious Stranger (1916) | 37 262 | 5580 |
| twa09 | A Fable (1909) | 810 | 307 |
| twa10 | Hunting the Deceitful Turkey (1906) | 1259 | 519 |
| twa11 | The McWilliamses And The Burglar Alarm (1882) | 2680 | 904 |
| twa12 | The Adventures of Tom Sawyer (1876) | 72 697 | 9996 |
| twa13 | Adventures of Huckleberry Finn (1884) | 114 973 | 9971 |
| twa14 | Tom Sawyer Abroad (1894) | 35 067 | 4676 |
| twa15 | Tom Sawyer, Detective (1896) | 24 078 | 3354 |
| H. G. Wells | | | |
| wel01 | The Time Machine (1895) | 32 391 | 5887 |
| wel02 | The Island of Dr. Moreau (1896) | 43 909 | 6696 |
| wel03 | The Wonderful Visit (1895) | 38 884 | 6709 |
| wel04 | The Wheels of Chance (1895/1935) | 55 824 | 9380 |
| wel05 | The Invisible Man (1897) | 49 460 | 7400 |
| wel06 | The War of the Worlds (1898) | 59 861 | 9063 |
| wel07 | The First Men in the Moon (1901) | 69 114 | 9266 |
| wel08 | The Passionate Friends (1913) | 103 694 | 12 852 |
| wel09 | The Shape of Things to Come (1933) | 156 204 | 18 662 |
| wel10 | The Soul of a Bishop (1917) | 80 080 | 11 066 |

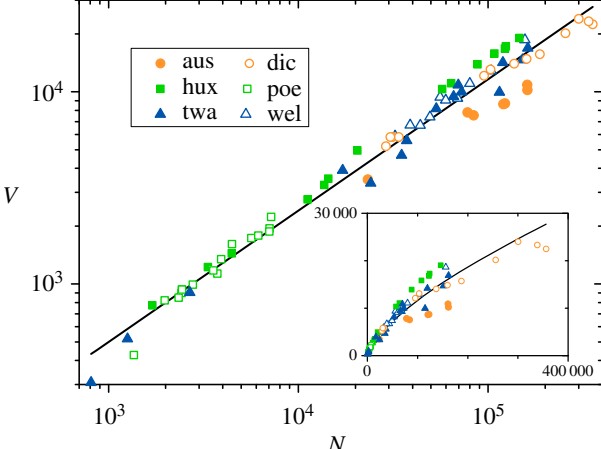

**Figure 1.** Heaps plot (vocabulary size $V$ versus text length $N$, measured in number of words) in log-log scales, for the 75 works in the corpus. Different symbols correspond to different authors (aus: J. Austen, dic: Ch. Dickens, hux: A. Huxley, poe: E. A. Poe, twa: M. Twain, wel: H. G. Wells; table 1). The inset shows the same data in linear-linear scales. Lines correspond to a power-law fitting, $V \propto N^h$, with $h = 0.68$.

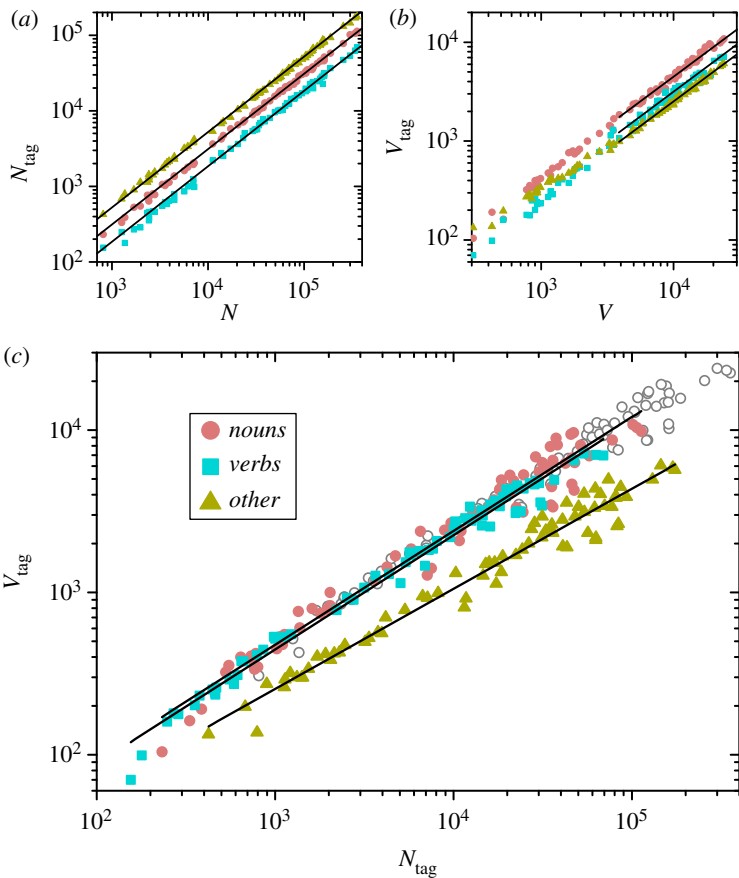

**Figure 2.** (*a*) Number of word tokens in each tagged class, $N_{tag}$ (tag = *nouns*, *verbs* and *others*) as a function of the text length $N$, in log-log scales, for the 75 works in the corpus. Straight lines have unitary slope. (*b*) As in (*a*), for the number of word types in each tag, $V_{tag}$, as a function of the vocabulary size $V$. (*c*) Heaps plot for the words in each tag, for the 75 works. Open symbols correspond the same data plotted in figure 1. The two upper straight lines, are fittings for *nouns* and *verbs*, both with slope $h_{n,v} = 0.70$. The lower straight line is the fitting for *others*, with slope $h_o = 0.62$.

Remarkably, for both *nouns* and *verbs*, the above proportionality constants satisfy the approximate relations $\beta_n \approx \alpha_n^h$ and $\beta_v \approx \alpha_v^h$, with $h$ the same exponent as obtained for the relation between $V$ and $N$. A direct consequence of these relations is that $V_n/V \approx (N_n/N)^h$ and $V_v/V \approx (N_v/N)^h$ which, in turn, implies that the log-log plots of $V_n$ versus $N_n$ and $V_v$ versus $N_v$ lie approximately over the same

straight line as $V$ versus $N$ in figure 1. This is clearly seen in figure 2*c*, where empty symbols stand for the data of figure 1. Linear fittings for *nouns* and *verbs*, shown as straight lines in figure 2*c*, have coincident slopes $h_{n,v} = 0.70 \pm 0.01$, with correlation coefficients $r = 0.98$ and $0.99$, respectively. For *others*, as demonstrated by the lowermost straight line, the relation between $V_o$ and $N_o$ is also well approximated by a power law, with exponent $h_o = 0.62 \pm 0.01$ and correlation coefficient $r = 0.98$.

In summary, interdependence between text lengths and vocabulary sizes for the 75 works in the corpus is in very good agreement with the power-law relation postulated by Heaps' Law, although systematic deviations from the average trend seem to occur for specific authors. The three grammatical classes considered here comprise well-defined fractions along each text and, for long texts, within each vocabulary. As for the relation between the number of word tokens and types in each tag, *nouns* and *verbs* closely follow the same power-law relation as the whole word collections, while *others* complies Heaps' Law with a smaller power-law exponent.

# 4. Quantifying the significance of Heaps functions

Turning the attention to the third variant of Heaps' Law referred to in the Introduction, we now consider the progressive appearance of new words along each individual text. As advanced in §1, for each text, we define the *Heaps function* $v(n)$ as the number of word types $v$ that have occurred up to the $n$-th word token (inclusive) along the whole text. For a text with a total length of $N$ word tokens and a vocabulary formed by $V$ word types, $v(n)$ is a non-decreasing function with $v(1) = 1$ and $v(N) = V$.

A straightforward test of statistical significance for the information provided by the Heaps function consists in comparing $v(n)$ for the text under study and for a shuffled version of the same text. More precisely, we can calculate the average of $v(n)$ over the whole set of different word orderings, $\bar{v}(n)$. Since, in this average, all possible orderings are equally represented, $\bar{v}(n)$ is solely determined by the numbers of occurrences, $m_1, m_2, \ldots, m_V$, of all the words in the vocabulary. Note that this set of numbers is equivalent to the information stored in Zipf's Law. In fact, if the vocabulary is ordered by decreasing number of occurrences, the graph of $m_r$ as a function of $r$ is nothing but Zipf's plot for the text in question.

In previous work on Heaps' Law, the average $\bar{v}(n)$ has been estimated numerically [13,22,23], as it seems to have passed unnoticed that its exact analytical expression has been available in the literature since at least four decades ago.[1] It reads [29]

$$\bar{v}(n) = \sum_{r=1}^{V} [1 - B_N(m_r, n)], \tag{4.1}$$

with

$$B_N(m_r, n) = \frac{\dbinom{N - m_r}{n}}{\dbinom{N}{n}}. \tag{4.2}$$

In this equation, the binomial coefficients are assumed to vanish,

$$\dbinom{k_1}{k_2} = 0, \tag{4.3}$$

if $k_2 > k_1$. The function $\bar{v}(n)$ grows monotonically with $n$ and, irrespective of the specific values $m_1, m_2, \ldots, m_V$, we have $\bar{v}(1) = 1$ and $\bar{v}(N) = V$.

The significance of the difference between the Heaps function for the actual text and $\bar{v}(n)$ can be assessed by comparison with the standard deviation over the different word orderings, $\sigma_v(n)$. The

---

[1]To the present authors' knowledge, the analytical expressions of equations (4.1) and (4.4) were first obtained in the framework of a traditional quantitative technique in ecology and other related life sciences, called *rarefaction* [29,30]. The basic problem is, given a large collection of objects divided into categories, to estimate the number of different categories obtained in a random extraction of a certain number of objects from that collection. In the original framework, objects and categories are—for instance—individuals and species in a collection of animals. In our problem, they respectively correspond to word tokens and types.

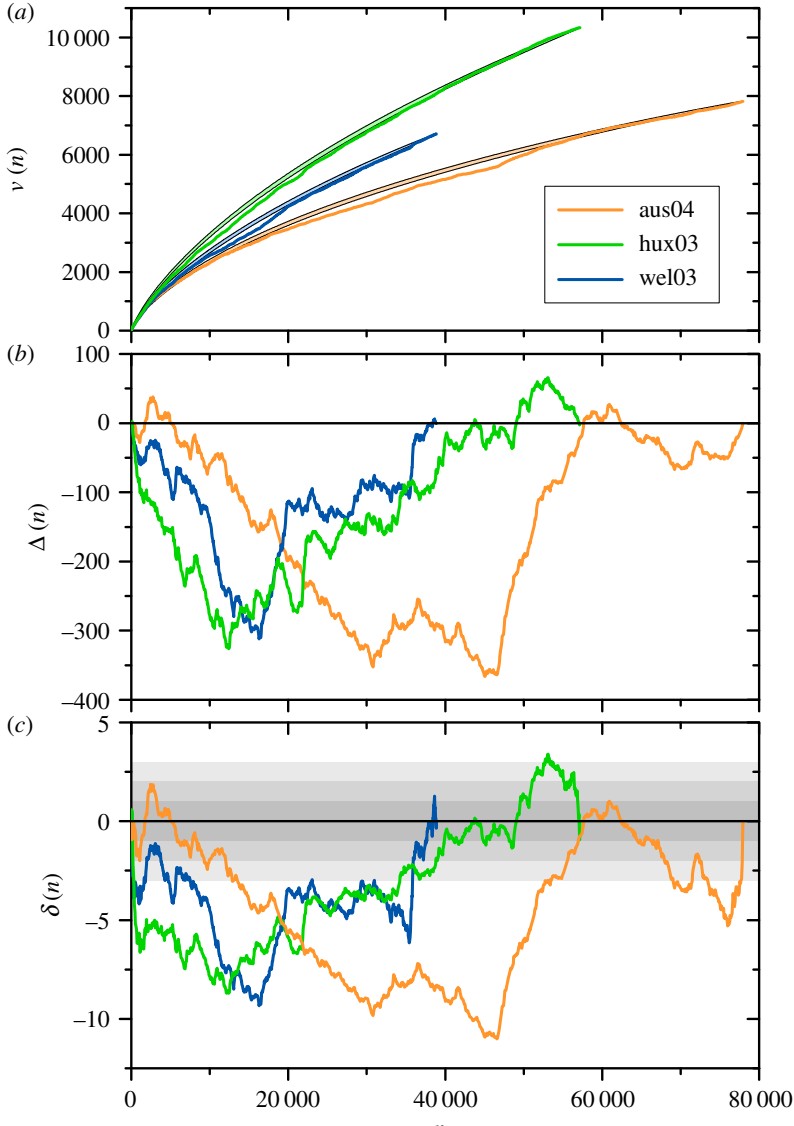

**Figure 3.** (a) Curves stand for the Heaps functions $v(n)$ of three works in the corpus, namely, Austen's *Northanger Abbey* (aus04), Huxley's *Chrome Yellow* (hux03) and Wells' *The Wonderful Visit* (wel03). Narrow shaded areas are bounded by the average functions $\bar{v}(n) \pm \sigma_v(n)$. (b,c) Respectively, the absolute and relative Heaps anomalies, defined as in equation (4.5), for the same three texts. Horizontal bands in (c) have integer widths, helping to appraise the absolute anomaly with respect to the standard deviation of randomized shufflings of the texts.

corresponding variance, whose exact analytical expression is also known [29], reads

$$\sigma_v^2(n) = \sum_{r=1}^{V} B_N(m_r, n)[1 - B_N(m_r, n)] + 2 \sum_{r=2}^{V} \sum_{s=1}^{r-1} [B_N(m_r + m_s, n) - B_N(m_r, n)B_N(m_s, n)], \qquad (4.4)$$

and satisfies $\sigma_v^2(1) = \sigma_v^2(N) = 0$. The value of $n$ for which $\sigma_v^2(n)$ attains its maximum, as well as the maximal value of $\sigma_v^2(n)$, depend on the specific set $m_1, m_2, \ldots, m_V$.

Figure 3a shows a comparison between the actual Heaps function for three works in the present corpus, and the respective averages and standard deviations. For each work, $v(n)$ is plotted as a curve. The analytical values obtained from equations (4.1) and (4.4) are represented as narrow shaded areas, limited above and below by the curves $\bar{v}(n) + \sigma_v(n)$ and $\bar{v}(n) - \sigma_v(n)$, respectively. These plots make it clear that $v(n)$ and $\bar{v}(n)$ are quite close to each other, although $v(n)$ is sometimes appreciably outside the corresponding shaded area. The likeness between $v(n)$ and $\bar{v}(n)$, which is verified for all the works in the present corpus—and which has been previously reported for a few other individual texts [13,23]—indicates, in particular, that $v(n)$ should not be expected to verify a Heaps-like law, $v \propto n^h$,

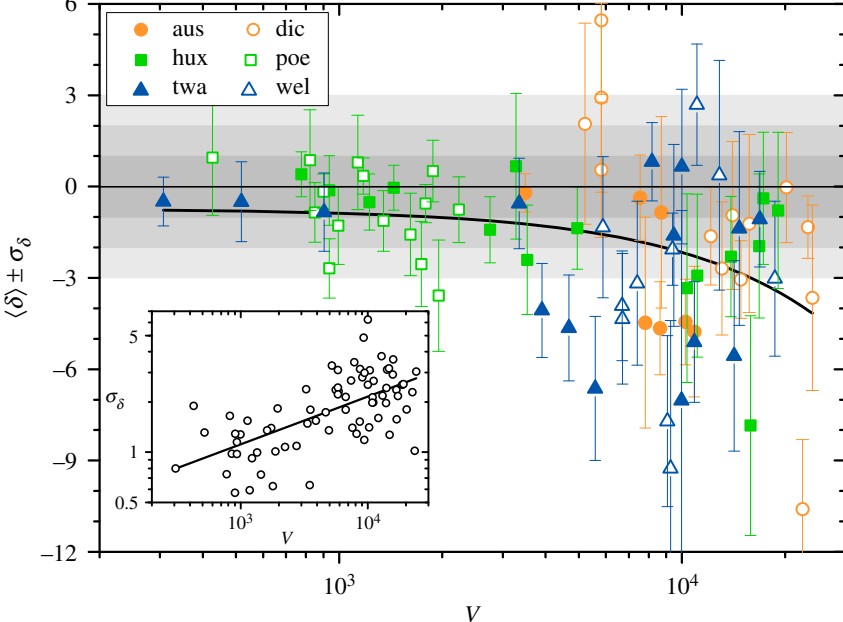

**Figure 4.** The relative anomaly averaged along each whole text, $\langle\delta\rangle$ (symbols) and the corresponding standard deviations $\sigma_\delta$ (error bars) for the 75 works in the corpus, as functions of their vocabulary sizes $V$. The horizontal axis is logarithmic, to ease discerning data in the low-$V$ zone. Different symbols correspond to different authors (figure 1). Horizontal shaded bands have integer widths. The curve corresponds to a linear fitting of $\langle\delta\rangle$ versus $V$, as described in the text. The inset shows a log-log plot of $\sigma_\delta$ versus $V$. The straight line has slope $s = 0.29$.

since the functional form of the average $\bar{v}(n)$, as given by equation (4.1), is not well described by a power-law approximation over any significantly long interval. This is consistent with previous reports on the Heaps function for individual texts [9,12,13,23], where the merest inspection reveals systematic deviations from a power-law interdependence.

To quantitatively compare the Heaps function $v(n)$ with the corresponding average $\bar{v}(n)$, we define the absolute and relative *Heaps anomalies* as

$$\Delta(n) = v(n) - \bar{v}(n) \quad \text{and} \quad \delta(n) = \frac{\Delta(n)}{\sigma_v(n)}, \tag{4.5}$$

respectively. Note that $\Delta(1) = \Delta(N) = 0$, while $\delta(n)$ is undefined at the two ends. Figure 3b,c shows the Heaps anomalies $\Delta(n)$ and $\delta(n)$ for the three works in figure 3a. The widths of the horizontal shaded bands in the lower panel correspond to integer values of $\delta(n)$, in order to graphically contrast the difference $v(n) - \bar{v}(n)$ with the standard deviation $\sigma_v(n)$. In the three cases, the difference reaches values which are, in modulus, between 8 and 12 times larger than the standard deviation, indicating a statistically highly significant deviation of the actual Heaps functions from the respective averages.

More strikingly, we see that for the three works considered in figure 3, both $\Delta(n)$ and $\delta(n)$ are predominantly negative. This regularity, which indicates that in the actual texts the appearance of new word types is for the most part retarded with respect to the average over word shufflings, turns out to be a widespread rule over the whole corpus, especially for long works. To demonstrate this feature in a compact way, we have calculated, for each work, the average and the variance of the relative anomaly along the text

$$\langle\delta\rangle = \frac{1}{N}\sum_{n=1}^{N}\delta(n) \quad \text{and} \quad \sigma_\delta^2 = \frac{1}{N}\sum_{n=1}^{N}(\delta(n) - \langle\delta\rangle)^2. \tag{4.6}$$

Symbols in figure 4 stand for the values of $\langle\delta\rangle$ as functions of the vocabulary size $V$ for the 75 works of the corpus. Error bars indicate the standard deviation $\sigma_\delta$. The curve corresponds to a linear regression between $\langle\delta\rangle$ and $V$ (note that in this plot the horizontal scale is logarithmic). Along this fitting, whose correlation coefficient is $r = -0.33$, not only is $\langle\delta\rangle$ always negative, but its modulus increases with the vocabulary size, indicating that the relative Heaps anomaly grows for longer texts. The inset shows

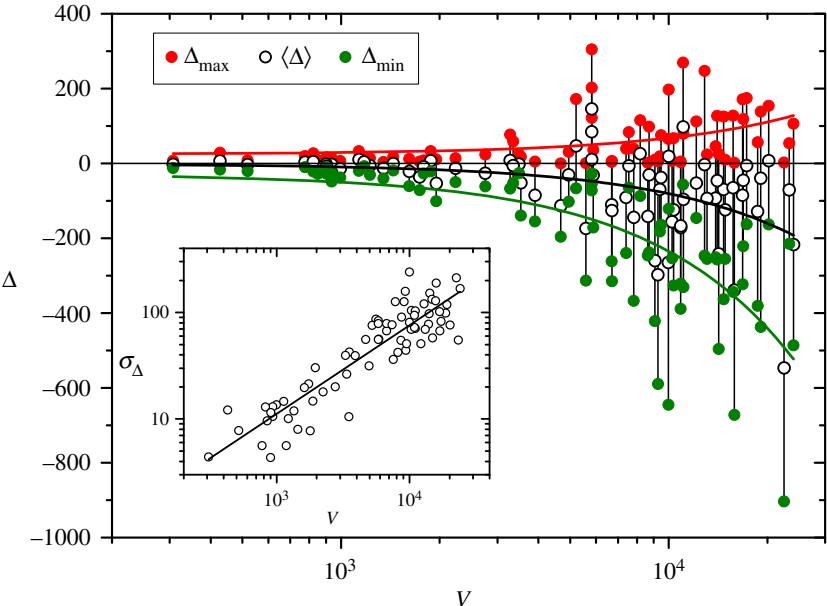

**Figure 5.** Maximum, average and minimum absolute anomaly—respectively, $\Delta_{max}$, $\langle\Delta\rangle$ and $\Delta_{max}$—along each of the 75 works in the corpus, as functions of their vocabulary sizes $V$. Curves correspond to linear fittings of the three quantities versus $V$. The inset shows a log-log plot of the standard deviation $\sigma_\Delta$ versus $V$, with $\sigma_\Delta$ computed along each work. The straight line has slope $s = 0.83$.

the individual standard deviations $\sigma_\delta$ as functions of $V$ in log-log scales. The linear fitting, corresponding to a power-law $\sigma_\delta \propto V^s$ with exponent $s = 0.29 \pm 0.04$, has correlation coefficient $r = 0.63$.

The above results on the relative Heaps anomaly averaged along each work clearly show that, generally, the difference between Heaps curves for actual texts and their shuffled versions has a substantial statistical significance. This significance, moreover, grows for longer works with richer vocabularies. For texts with vocabularies of $10^4$ words and beyond, the difference can be some 10 times larger than the deviation expected by chance.

We now turn the attention to the analysis of the absolute anomaly $\Delta$ which, as shown below, allows for a more straightforward comparison between actual and random word orderings when it comes to tagged texts. Empty symbols in the main panel of figure 5 correspond to the absolute anomaly of each work averaged along the whole text, $\langle\Delta\rangle$, as a function of the vocabulary size. Joined by a vertical line to each empty symbol, full symbols show the overall maximum $\Delta_{max}$ and minimum $\Delta_{min}$ along each text. Curves stand for linear fittings of each one of the three sets, with correlation coefficients $r = 0.39$, $-0.48$ and $-0.71$ for $\Delta_{max}$, $\langle\Delta\rangle$ and $\Delta_{min}$, respectively. As expected from the results shown in figure 4, the average trend is that $\langle\Delta\rangle$ remains negative, with absolute values increasing as the vocabulary grows, confirming that the introduction of new word types is on the average retarded with respect to random word orderings. Note that $\Delta_{min}$ can reach negative values of several hundreds for the largest vocabularies. The inset of figure 5 shows the standard deviation of the absolute anomaly as a function of the vocabulary size, in log-log scales. These data admit a sharper linear fitting than for the relative anomaly (see inset of figure 4), with slope $s = 0.83 \pm 0.05$ and correlation coefficient $r = 0.90$.

## 5. Discerning between grammatical classes

Focusing on the contribution of each tagged class to the absolute Heaps anomaly $\Delta$ considered in the preceding section, we first note that the Heaps function $v(n)$ can be straightforwardly divided into three terms, $v(n) = v_n(n) + v_v(n) + v_o(n)$, where $v_{tag}(n)$ (tag = *nouns*, *verbs* and *others*) indicates the total number of occurrences of each tag up to the $n$-th word (inclusive) along the whole text. A measure of the contribution of each tag to $v(n)$, by comparison to a random distribution of words along the text, is given by the *Heaps excess*

$$E_{tag}(n) = v_{tag}(n) - \frac{V_{tag}}{V} v(n), \tag{5.1}$$

where the ratio $V_{tag}/V$ represents the fraction of each tag in the vocabulary (cf. figure 2b). If the words in each tag were uniformly distributed all along each text, we would expect $E_n(n) \approx E_v(n) \approx E_o(n) \approx 0$ for all $n$.

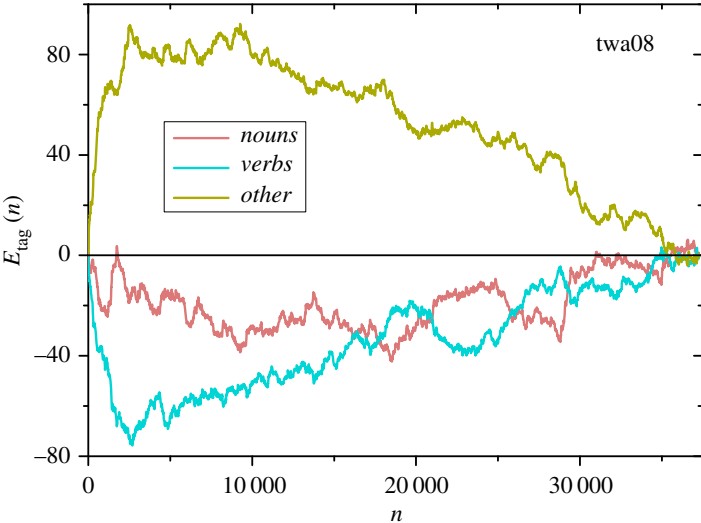

**Figure 6.** The Heaps excess $E_{\text{tag}}$ as a function of $n$ and for each tag, as defined by equations (5.1) and (5.3), along Twain's *The Mysterious Stranger* (twa08, $N = 37262$, $V = 5580$).

Conversely, a systematic deviation from zero would indicate persistent heterogeneity in the distribution of the corresponding words. From equation (5.1), moreover, we have $E_n(n) + E_v(n) + E_o(n) = 0$ for all $n$. By construction, therefore, a positive or negative excess in a tag must necessarily be balanced by an excess of the opposite sign in at least one of the other tags. In this sense, the quantities $E_{\text{tag}}(n)$ measure distribution deviations of tags respective to each other.

Note also that, defining the absolute Heaps anomaly of each tag as

$$\Delta_{\text{tag}}(n) = v_{\text{tag}}(n) - \frac{V_{\text{tag}}}{V}\bar{v}(n), \tag{5.2}$$

the Heaps excess introduced in equation (5.1) can be rewritten as

$$E_{\text{tag}}(n) = \Delta_{\text{tag}}(n) - \frac{V_{\text{tag}}}{V}\Delta(n); \tag{5.3}$$

cf. equation (4.5). Comparing this expression with equation (5.1), it becomes clear that the Heaps excess, as a measure of the heterogeneity in the distribution of each tag along the text, can also be interpreted in terms of its contribution to the absolute anomaly. If all tags would uniformly add to $\Delta(n)$, we should expect $E_{\text{tag}}(n) \approx 0$ for all $n$. Significant values of the Heaps excess can therefore be interpreted as deviations with respect to tag homogeneity in $\Delta(n)$.

Figure 6 shows the Heaps excess $E_{\text{tag}}(n)$ for the tree tags in an individual work of the corpus, namely, Twain's *The Mysterious Stranger* (twa08). This example illustrates a trend that is found for other works, as we show below. Specifically, the tag *others* typically exhibits rather large, positive values of $E_o(n)$, compensated by smaller, negative values of $E_n(n)$ and $E_v(n)$, for *nouns* and *verbs*, respectively. Among the two latter, moreover, the Heaps excess for *verbs* is on the average more negative than for *nouns*.

Figure 7a–c shows plots similar to that of figure 5, now with the Heaps excess as a function of the number of word types in each tag, $V_{\text{tag}}$, for the 75 works of the corpus. For each work, the middle dot stands for the average of $E_{\text{tag}}(n)$ over the whole text. The upper and the lower dots, in turn, represent the maximal and minimal values attained by $E_{\text{tag}}(n)$. In the respective insets, whose horizontal axes coincide with those of the main plots, we show the standard deviations of the Heaps excess along each work.

Comparison of the panels reveals substantial differences in the behaviour of $E_{\text{tag}}$ for the three tagged classes. In figure 7a, we see that the distribution of maximal, average, and minimal values of the Heaps excess for *nouns* is markedly symmetric around zero. The linear fitting of the average, with correlation coefficient $r = 0.06$, is barely distinguishable from the horizontal line at $E_n = 0$. In turn, linear fittings for maxima and minima are virtually symmetric to each other. Their respective correlation coefficients are $r = 0.63$ and $-0.61$. As for the standard deviation, it is well approximated ($r = 0.92$) by a power law $\sigma_E \propto V_n^s$, with $s = 0.66 \pm 0.03$.

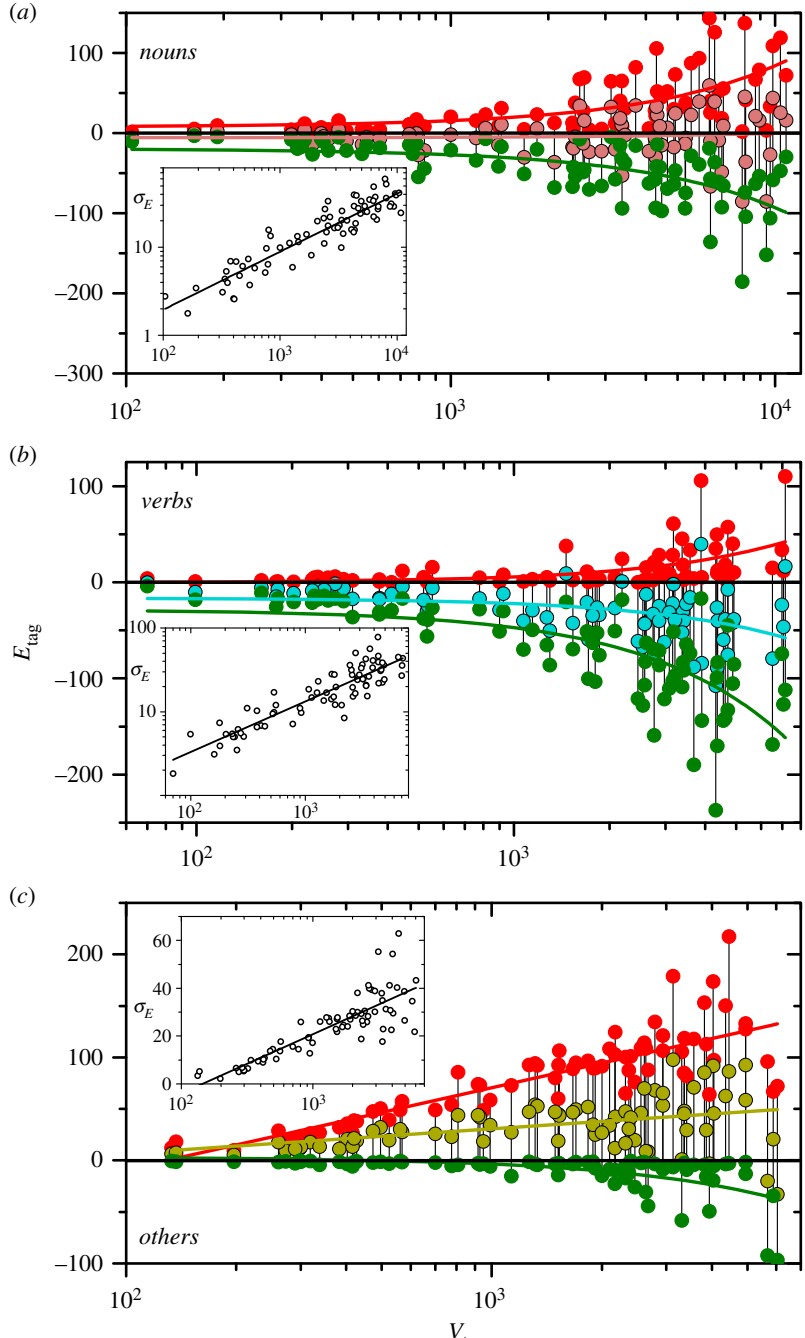

**Figure 7.** (a) Maximum, average and minimum Heaps excess for *nouns* along each of the 75 works in the corpus, as functions of the number of nouns in the vocabularies, $V_n$. Curves correspond to linear fittings. The inset shows the Heap excess standard deviation along each text. (b) Same as (a), for *verbs*. (c) Same as (a), for *others*. For the average and maximum Heaps excess, the fittings are here linear versus the logarithm of the vocabulary size. Note that, in contrast with the other panels, the inset is plotted in linear-log scales. For clarity, the axes labels have been indicated only once, but they are the same in all plots.

On the other hand, as shown in the middle panel, *verbs* are clearly biased towards negative values. The linear fitting for the average, with $r = -0.41$, reaches values below $E_v = -50$ for the largest vocabularies. Meanwhile, the minimal excesses for vocabularies larger than a few thousands lie typically between $-200$ and $-100$. By contrast, maximal excesses are consistently positive and, except for a few cases, they are always below 50. Correlation coefficients for the linear fittings of maxima and minima are $r = 0.52$ and $-0.71$, respectively. In the inset, the linear fitting for the standard deviation, with $r = 0.90$, corresponds to a power law with slope $s = 0.60 \pm 0.03$.

Finally, we see from figure 7c that the distribution of the Heaps excess for *others* is roughly symmetric to that of *verbs* with respect to $E_{tag} = 0$. In fact, while the minima of $E_o$ are negative and very close to zero,

maxima are always positive and grow with the vocabulary size reaching values between 100 and 200 for the largest vocabularies. This overall symmetry between *verbs* and *others* is, on the whole, not unexpected, because of the symmetric distribution of the Heaps excess for *nouns* (figure 7a) and the fact that the sum of the excesses for the three tags must vanish at any point along the text.

Closer inspection of the Heaps excess for *others*, however, reveals an important difference with that of *verbs*, regarding its dependence on the vocabulary size. While averages and minima of $E_v$ are reasonably well approximated by linear fittings—as shown by the curves in the middle panel, with the correlations reported in the preceding paragraph—averages and maxima of $E_o$ exhibit systematic deviations from a linear trend. The linear-log scales of figure 7, in fact, clearly suggest that a much better fitting for both averages and maxima is a linear interdependence between $E_o$ and $\ln V_o$. The straight lines correspond to fittings with correlation coefficients $r = 0.42$ and 0.83, respectively, for averages and maxima, while linear fittings in linear-linear scales produce substantially lower correlations. This distinctive behaviour of the Heaps excess for *others* appears also in its standard deviation. Note that, in contrast with *verbs* and *nouns*, the inset plot has linear-log scales. The linear fitting, with $r = 0.84$, again suggests a dependence on the vocabulary size of the type $\sigma_E \propto \ln V_o$.

The fact that the Heaps excess of *nouns* remains relatively close to zero—even for long texts—indicates that the anomaly in the appearance of new word types of this specific tag follows, on the average, the same trend as for the overall vocabulary. As we have seen in §4, such trend amounts to a systematic retardation in drawing upon the available vocabulary as compared with the average of random shufflings of the text. The predominantly negative excess for *verbs*, in turn, shows that the contribution of this tag to the anomaly is itself retarded with respect to the expected average. Compensating this tendency, the mostly positive values of the Heaps excess for *others* reveal a comparatively early appearance of new word types belonging to this tag.

# 6. Discussion

In this closing section, we briefly comment on the results that, in our view, are the most relevant contributions of Heaps analysis to the understanding of statistical patterns of language in the texts of the studied corpus. Although we are not able to provide an explanation of those results within a formal linguistic framework, we find it possible to point out several regularities that may provide useful insight on structural features in the production of written language, in particular, connecting texts and vocabularies.

In the first place, we mention the distinct overall participation of the tag *others* as compared with that of *nouns* and *verbs* (figure 2). The last two represent well-defined proportions of texts and vocabularies, with consistently more *nouns* than *verbs*. Meanwhile, the tag *others* constitutes the largest fraction of texts, but the smallest fraction of vocabularies—at least, when the vocabularies are large. For smaller vocabularies, on the other hand, the representation of *others* grows. This may be due to the fact that this tag comprises function words (or 'stop words,' such as conjunctions, connectors, articles, etc.) whose use is hardly avoidable in any text, no matter how short. They are therefore always present in any vocabulary, and may be dominant when the vocabulary is small. As the vocabulary grows, such words as adjectives and adverbs, many of which are directly related to specific nouns and verbs, are better represented in *others*, and the participation of this tag becomes similar to that of *nouns* and *verbs*. The peculiar behaviour of *others* may also be underlying the different power-law relation between the number of word tokens and types for this tag with respect to that of the other two tags, which closely coincides with the relation for the whole texts.

From the analysis of the difference between the Heaps functions of individual texts and the average of text shufflings, the most intriguing observation is the consistent retardation in the appearance of new word types in real texts with respect to the corresponding averages (figures 3–5). This effect, which becomes more conspicuous in longer texts with richer vocabularies, seems to point out a generic global feature in the production of literary discourse, in the form of a sustained delay in the occurrence of the elements that progressively create and shape the semantic context of the work [31]. The delay is perhaps related to the need of establishing connections between already present elements, creating the word clusters that establish topicality [22] and 'networks of concepts' [32,33], before new elements are introduced. On the other hand, taking into account the distinct nature of the words in the tag *others* referred to in the preceding paragraph, it may be that the need of introducing function words to comply with grammatical rules, which uniformly apply from the beginning of the text, implies a relative retardation in the use of words with more lexical meaning, which appear when

specific semantic contexts are being built. The widely disparate contributions of the three tags to the anomaly in the appearance of new word types, quantified by their respective excesses (figures 6 and 7), represents a remarkable finding that could support this second alternative. In any case, discerning between the two possibilities—or detecting a combination of both—would require a more 'microscopic' analysis of the progressive use of new words, possibly discriminating between a larger set of grammatical classes.

The corpus studied in this contribution has been assembled having in mind not only a certain uniformity in language—namely, standard narrative English in a circumscribed historical period—but also a degree of homogeneity in style, choosing a set of genres that should ensure a well-developed, self-contained discourse along each individual text. The lengths spanned by the selected works are adequate representatives of the narrative style, from short fables to long novels. Although the corpus could be expanded in several directions, the consistency of the statistical regularities revealed by our analysis of these 75 works suggests that they stand for significant features in the usage of language. As such, they should find correspondences in other corpora. We have already studied a small collection of literary works written in different languages—namely, Latin, Spanish, German, Finnish and Tagalog, for which automated tagging techniques are not yet as developed as for English—and found that the phenomenon of relative retardation in the appearance of new words is also present in the longest works. This, however, should not come as a surprise, in view of the possibility of translating texts between those languages. Although translation is of course not a word-by-word process, a parallelism between the creation of semantic contents along the same work in different languages is expected to emerge over the scales relevant to the regularities disclosed by Heaps analysis.

The identification of statistically significant regularities in data corpora of broader origins may point to the usefulness of Heaps functions—and, in general, Heaps-like analyses—not only in the processing of natural languages but also in the characterization of other complex combinatorial structures [34], such as those created by generative models [35], evolution and learning (both natural and artificial [36]), as well as innovation mechanisms [15].

Data accessibility. Plain-text files containing the pre-processed texts of the 75 literary works analysed in this contribution and the computational codes are deposited in Dryad Digital Repository: https://doi.org/10.5061/dryad.d51c59zz7 [37].

Authors' contributions. A.C. and D.H.Z. conceived and designed the research. D.H.Z. collected and pre-processed the corpus texts. A.C. wrote the computational codes and performed the numerical analysis. A.C. and D.H.Z. interpreted the results and wrote the manuscript. Both authors gave final approval for publication and agreed to be accountable for all aspects of the work.

Competing interests. We declare we have no competing interests.

Funding. This work was financially supported by Consejo Nacional de Investigaciones Científicas y Técnicas, Argentina.

Acknowledgements. Both authors acknowledge financial support from Consejo Nacional de Investigaciones Científicas y Técnicas, Argentina.

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
