## [Reviewer comments · Royal Society Open Science]

Review History

RSOS-200008.R0 (Original submission)

Review form: Reviewer 1

Is the manuscript scientifically sound in its present form?

Yes

Are the interpretations and conclusions justified by the results?

Yes

Is the language acceptable?

Yes

Do you have any ethical concerns with this paper?

No

Have you any concerns about statistical analyses in this paper?

No

Recommendation?

Accept with minor revision (please list in comments)

Comments to the Author(s)

In this work the authors study statistical regularities in natural language. Specifically, they consider samples from written text and study the growth of the vocabulary (Heaps' law). Using analytical calculations and computational analysis of 75 different books, they find substantial differences between Heaps curves of different word classes (nouns, verbs, ...).

The latter aspect constitutes a new contribution to the analysis of Heaps' law in the context of statistical laws in natural language. The findings are substantiated by the statistical analysis and the employed methodology is sound. While the authors cannot provide an explanation for their (admittedly curious) findings, the empirical findings alone will serve as a starting point for future analysis. The manuscript is written very clearly and the careful selection of figures make it easy to follow the different steps in the analysis.

Therefore, I fully recommend publication of the current manuscript.

I would only ask for minor revision in terms of the description of the methodology in order to ensure the reproducibility of the analysis. First, the description of the pre-processing of the data seems insufficient. For example, when using the NLTK-tokenizer, did the authors filter any words? Second, I couldnt find any information on the repository where the code for the analysis of the data will be published (criteria for publication state that "Datasets, code, and other digital materials should be deposited in an appropriate, recognised, publicly available repository").

Review form: Reviewer 2

Is the manuscript scientifically sound in its present form?

Yes

Are the interpretations and conclusions justified by the results?

Yes

Is the language acceptable?

Yes

Do you have any ethical concerns with this paper?

No

Have you any concerns about statistical analyses in this paper?

No

Recommendation?

Accept with minor revision (please list in comments)

Comments to the Author(s)

This manuscript investigates Heaps' law in literary texts. The main contribution of this manuscript is the analysis of the words classified by different parts of speech. As far as I am aware this is the first manuscript that performs this analysis, which adds a meaningful contribution tot this area of study. The manuscript is clearly written and the statistical analysis, including the comparison to null models, are correctly performed. I recommend the manuscript for publication after the authors address the points listed below.

1) I found the second sentence of the abstract unclear, and the whole abstract over complicated. I suggest trying to simplify this sentence and to focus on the main results.

2) It would be helpful if the authors would use the distinction between word types and word tokens, which is standard in linguistics. For instance, in the caption of Fig. 2 it is sometimes hard to distinguish which concept is being referred (Also in other parts of the manuscript).

3) The main result of the manuscript in Fig. 2 is very interesting. For large values of N_{tag} the newly added V_{tag} must correspond to very rare words, possibly including words that are not in standard dictionaries or list of words. These results rely heavily on the POS tagger and it'd be in general important to add more information about how it is tagging the words. It is remarkable that the POS tagger seem to consistently tag these words as nouns, verbs, and others. It'd be helpful if the authors would add some information about how the POS tagger works and whether it can be trusted even for extremely rare words? Should the scaling be expected even for $N \rightarrow \infty$ and is the tagging reliable in this limit?

4) In Sec. 4 and in the third paragraph of the discussion 6 the authors discuss the comparison to randomized texts. This is an important part of the manuscript, which is indeed very relevant and contains original contribution. However, I believe that the observation that randomized texts show larger V_{tags} is not new. It has been observed and derived mathematically in Ref. 22, but probably it was known even earlier. It is natural to expect taking into account that the usage of words in the text is correlated, with words clustering in regions of the text. This correlation delays the appearance of new words. It is not present in shuffled text, which therefore have a larger number of distinct word tokens for texts of similar length. I believe the authors should revise their claims of novelty in this aspect, specially in the Discussion.

5) All data is available and the Dryad server is working. However, it is not clear how to replicate the results because there is no code available for the data filtering or for the NLP part. The books in the repository still contain metadata, such as translator's preface, and it is not clear how the authors performed the filtering (if any).

Decision letter (RSOS-200008.R0)

12-Feb-2020

Dear Dr Zanette,

On behalf of the Editors, I am pleased to inform you that your Manuscript RSOS-200008 entitled "Heaps' law and Heaps functions in tagged texts: Evidences of their linguistic relevance" has been accepted for publication in Royal Society Open Science subject to minor revision in accordance with the referee suggestions. Please find the referees' comments at the end of this email.

The reviewers and handling editors have recommended publication, but also suggest some minor revisions to your manuscript. Therefore, I invite you to respond to the comments and revise your manuscript.

- Ethics statement

- Data accessibility

It is a condition of publication that all supporting data are made available either as supplementary information or preferably in a suitable permanent repository. The data

accessibility section should state where the article's supporting data can be accessed. This section should also include details, where possible of where to access other relevant research materials such as statistical tools, protocols, software etc can be accessed. If the data has been deposited in an external repository this section should list the database, accession number and link to the DOI for all data from the article that has been made publicly available. Data sets that have been deposited in an external repository and have a DOI should also be appropriately cited in the manuscript and included in the reference list.

If you wish to submit your supporting data or code to Dryad (<http://datadryad.org/>), or modify your current submission to dryad, please use the following link:
<http://datadryad.org/submit?journalID=RSOS&manu=RSOS-200008>

- **Competing interests**

- **Authors' contributions**

- **Acknowledgements**

- **Funding statement**

Because the schedule for publication is very tight, it is a condition of publication that you submit the revised version of your manuscript before 21-Feb-2020. Please note that the revision deadline will expire at 00.00am on this date. If you do not think you will be able to meet this date please let me know immediately.

If your manuscript is newly submitted and subsequently accepted for publication, you will be asked to pay the article processing charge, unless you request a waiver and this is approved by Royal Society Publishing. You can find out more about the charges at <https://royalsocietypublishing.org/rsos/charges>. Should you have any queries, please contact openscience@royalsociety.org.

Kind regards,
Lianne Parkhouse
Editorial Coordinator

on behalf of Professor Matjaz Perc (Associate Editor) and Mark Chaplain (Subject Editor)
openscience@royalsociety.org

Reviewer comments to Author:

Reviewer: 1

Comments to the Author(s)

In this work the authors study statistical regularities in natural language. Specifically, they consider samples from written text and study the growth of the vocabulary (Heaps' law). Using analytical calculations and computational analysis of 75 different books, they find substantial differences between Heaps curves of different word classes (nouns, verbs, ...).

The latter aspect constitutes a new contribution to the analysis of Heaps' law in the context of statistical laws in natural language. The findings are substantiated by the statistical analysis and the employed methodology is sound. While the authors cannot provide an explanation for their (admittedly curious) findings, the empirical findings alone will serve as a starting point for future analysis. The manuscript is written very clearly and the careful selection of figures make it easy to follow the different steps in the analysis.

Therefore, I fully recommend publication of the current manuscript.

I would only ask for minor revision in terms of the description of the methodology in order to ensure the reproducibility of the analysis. First, the description of the pre-processing of the data seems insufficient. For example, when using the NLTK-tokenizer, did the authors filter any words? Second, I couldnt find any information on the repository where the code for the analysis of the data will be published (criteria for publication state that "Datasets, code, and other digital materials should be deposited in an appropriate, recognised, publicly available repository").

Reviewer: 2

Comments to the Author(s)

This manuscript investigates Heaps' law in literary texts. The main contribution of this manuscript is the analysis of the words classified by different parts of speech. As far as I am aware this is the first manuscript that performs this analysis, which adds a meaningful contribution tot this area of study. The manuscript is clearly written and the statistical analysis, including the comparison to null models, are correctly performed. I recommend the manuscript for publication after the authors address the points listed below.

1) I found the second sentence of the abstract unclear, and the whole abstract over complicated. I suggest trying to simplify this sentence and to focus on the main results.

2) It would be helpful if the authors would use the distinction between word types and word tokens, which is standard in linguistics. For instance, in the caption of Fig. 2 it is sometimes hard to distinguish which concept is being referred (Also in other parts of the manuscript).

3) The main result of the manuscript in Fig. 2 is very interesting. For large values of N_{tag} the newly added V_{tag} must correspond to very rare words, possibly including words that are not in standard dictionaries or list of words. These results rely heavily on the POS tagger and it'd be in general important to add more information about how it is tagging the words. It is remarkable

that the POS tagger seem to consistently tag these words as nouns, verbs, and others. It'd be helpful if the authors would add some information about how the POS tagger works and whether it can be trusted even for extremely rare words? Should the scaling be expected even for $N \rightarrow \infty$ and is the tagging reliable in this limit?

4) In Sec. 4 and in the third paragraph of the discussion 6 the authors discuss the comparison to randomized texts. This is an important part of the manuscript, which is indeed very relevant and contains original contribution. However, I believe that the observation that randomized texts show larger V_{tags} is not new. It has been observed and derived mathematically in Ref. 22, but probably it was known even earlier. It is natural to expect taking into account that the usage of words in the text is correlated, with words clustering in regions of the text. This correlation delays the appearance of new words. It is not present in shuffled text, which therefore have a larger number of distinct word tokens for texts of similar length. I believe the authors should revise their claims of novelty in this aspect, specially in the Discussion.

5) All data is available and the Dryad server is working. However, it is not clear how to replicate the results because there is no code available for the data filtering or for the NLP part. The books in the repository still contain metada, such as translator's preface, and it is not clear how the authors performed the filtering (if any).

Author's Response to Decision Letter for (RSOS-200008.R0)

See Appendix A.

Decision letter (RSOS-200008.R1)

21-Feb-2020

Dear Dr Zanette,

It is a pleasure to accept your manuscript entitled "Heaps' law and Heaps functions in tagged texts: Evidences of their linguistic relevance" in its current form for publication in Royal Society Open Science. The comments of the reviewer(s) who reviewed your manuscript are included at the foot of this letter.

on behalf of Professor Matjaz Perc (Associate Editor) and Mark Chaplain (Subject Editor)
openscience@royalsociety.org

Appendix A

RESPONSE TO REVIEWERS - Manuscript ID RSOS-200008

REVIEWER 1

> First, the description of the pre-processing of the data seems insufficient. For example, when using the NLTK-tokenizer, did the authors filter any words?

Response: We are now more specific in section 2 on how the analyzed texts are pre-processed and give some details on the tokenization and tagging algorithms. In particular, tokenization does not use any kind of word filtering. In any case, full documentation of the NLTK library is available in Refs. 25 and 26, and a related reference (now Ref. 27) has been added.

> I couldnt find any information on the repository where the code for the analysis of the data will be published (criteria for publication state that "Datasets, code, and other digital materials should be deposited in an appropriate, recognised, publicly available repository")

Response: Computational codes have now been made available in Dryad.

REVIEWER 2

> 1) I found the second sentence of the abstract unclear, and the whole abstract over complicated. I suggest trying to simplify this sentence and to focus on the main results.

Response: We have rephrased several sentences in the abstract, emphasizing our conclusions. We hope that the text is now clearer.

> 2) It would be helpful if the authors would use the distinction between word types and word tokens, which is standard in linguistics. For instance, in the caption of Fig. 2 it is sometimes hard to distinguish which concept is being referred (Also in other parts of the manuscript).

Response: Following this suggestion, we now use the nomenclature of "word tokens" and "word types" along the text. For the convenience of the reader who may be not familiar with it, we define word tokens and types the first time they appear, in the second paragraph of the introduction.

> 3) It'd be helpful if the authors would add some information about how the POS tagger works and whether it can be trusted even for extremely rare words? Should the scaling be expected even for $N \rightarrow \infty$ and is the tagging reliable in this limit?

Response: In section 2, we now give more details on the NLTK tagger (see first response to Reviewer 1). Although there are no specifications on how "extremely rare" a word has to be for the method to fail, we have not been able to detect any gross error in the tagging process. Although it might be an interesting question from a formal viewpoint, considering the limit $N \rightarrow \infty$ makes little sense in our analysis, as unitary English texts spanning more than a few hundred thousand word tokens

(like some of the works considered here, cf. dic09, dic10) are, to our knowledge, inexistent.

> 4) I believe that the observation that randomized texts show larger V_tags is not new. It has been observed and derived mathematically in Ref. 22, but probably it was known even earlier. It is natural to expect taking into account that the usage of words in the text is correlated, with words clustering in regions of the text. This correlation delays the appearance of new words. It is not present in shuffled text, which therefore have a larger number of distinct word tokens for texts of similar length. I believe the authors should revise their claims of novelty in this aspect, specially in the Discussion.

Response: The corpora studied in Ref. 22 consist of collections of untagged texts of different origins, and the analysis of the vocabulary size as a function of text length is not done -as is our case- along each individual text (i.e. as the text progresses) but for the total vocabulary and length of each text (i.e. across texts; see their section 3). However, their discussion on word clustering and topicality is relevant to our comment on the possible causes of retardation in the appearance of new word types in real texts. We have therefore lengthened this comment in section 6, and cited the reference again.

> 5) All data is available and the Dryad server is working. However, it is not clear how to replicate the results because there is no code available for the data filtering or for the NLP part. The books in the repository still contain metadata, such as translator's preface, and it is not clear how the authors performed the filtering (if any).

Response: Computational codes have now been made available in Dryad.